# Differences in auditory brainstem responses between laboratory-reared and wild-caught prairie voles (*Microtus ochrogaster*)

**Luberson Joseph, Naleyshka Colon-Rivera, Emily M. New, Desi M. Joseph, Jessica A. Hurd, Casey E. Sergott, Elizabeth A. McCullagh**◉*

Department of Biology, College of Arts and Sciences, Oklahoma State University, Stillwater, Oklahoma, United States of America

* elizabeth.mccullagh@okstate.edu

## Abstract

Prairie voles (*Microtus ochrogaster*) are a semi-fossorial rodent that are an emerging model in social neuroscience. Comparing laboratory-reared and wild-caught individuals is essential for understanding how environmental history shapes neural and sensory traits and for assessing the ecological validity of laboratory findings. Despite this, relatively few studies have taken this approach. We used auditory brainstem responses (ABRs) to compare ABR thresholds and ABR wave characteristics between laboratory-reared and wild-caught prairie voles. ABR recordings show that, similar to other semi-fossorial rodents, *M. ochrogaster* exhibit a hearing range of 1–46 kHz with peak sensitivity between 8–24 kHz in wild-caught and 8–32 kHz in laboratory-reared voles. However, wild-caught prairie voles displayed significantly lower ABR thresholds at 1, 4, 8, 16, and 24 kHz compared to laboratory-reared prairie voles. There were significant differences in interpeak latency between both tested groups, with laboratory-reared prairie voles showing faster interpeak latency responses than wild-caught voles. However, there were no differences in amplitude ratios between groups. Laboratory-reared prairie voles showed faster normalized latencies and higher relative amplitude of the binaural interaction component (BIC) of the ABR than wild-caught voles. There were no significant differences in ABR thresholds, interpeak latency, amplitude ratio, normalized latency, and relative amplitude between the sexes. These differences in auditory processing support the importance of integrating both wild and captive populations to advance comparative auditory research.

## Introduction

Rodents represent the most species-rich order of mammals and make up nearly half of all the mammalian taxa described. They inhabit diverse habitats, from arid deserts

**Data availability statement:** Figshare doi: 10.6084/m9.figshare.30898388.

**Funding:** This work was supported by the National Science Foundation (2440070 to EAM).

**Competing interests:** The authors have declared that no competing interests exist.

to dense forests, and exhibit substantial diversity in their acoustic communication systems [1–3]. To date, extensive laboratory studies have been focused on rodents' auditory perception, most commonly for laboratory-reared mice (*Mus musculus*) and rats (*Rattus norvegicus domestica*), which represent non-human mammalian models in the auditory and other medically related fields [4–6]. In the laboratory, measuring auditory perception across strains of mice and rats has proven to be critical in pinpointing modifier genes [7,8], characterizing mouse strains [9], localizing and identifying gene mutations [10], and understanding the genetic and molecular mechanisms underlying auditory function and hearing loss [11,12].

Despite a substantial number of studies on laboratory-reared rodents' auditory perception [13–15] we still lack an understanding of the auditory abilities of their wild counterparts. This gap is important because wild animals are exposed to variable acoustic environments and natural selection pressures that may shape auditory processing differently from controlled laboratory settings [16]. While research devoted to laboratory-reared rodents has aided in discovering auditory mechanisms and responses [13,14], it is essential to continue investigating hearing in wild animals. Such studies could share novel insights on the specialized hearing adaptivity of rodents, including features important for maintaining social bonds, conspecific communication and recognition, finding mates and food sources, avoiding predators, and navigating noisy and complex acoustic landscapes [17]. Also, such efforts could significantly improve research in the auditory field and facilitate comparison studies between wild and laboratory-reared rodents, potentially addressing limitations associated with reduced genetic diversity and constrained natural variability in laboratory-based studies [18,19].

Most of our understanding of mammalian vocal communication is based on in-depth studies in inbred, laboratory-reared rodents, however, it remains unclear whether these findings accurately reflect the behaviors of free-living outbred populations. Notably, despite overall similarities in vocal characteristics (e.g., number of syllables and bandwidth duration), wild California mice (*Peromyscus californicus*) produce USVs with longer syllables (1SVs and 3SVs), emit calls at higher frequencies, and exhibit greater variability in USV frequencies compared to their laboratory-reared counterparts [18]. These vocal differences suggest that auditory sensitivity may also differ between wild and laboratory-reared populations, particularly given the co-evolution of vocal production and auditory perception [20,21]. It is therefore essential to compare auditory sensitivity across wild and laboratory rodents to gain a more comprehensive understanding of differences that might exist in auditory processing between these groups.

Prairie voles, *Microtus ochrogaster*, (family Cricetidae) are a semi-fossorial rodent that are native to central and eastern North America [22]. They are a socially monogamous rodent that form enduring pair bonds in which both males and females display parental care, engage in brooding, nest construction, grooming, and territory defense [23,24]. For the past few decades, prairie voles have emerged as a valuable model species in social neuroscience owing to their pair bonding and alloparenting behavior [25,26]. Indeed, previous studies involving prairie voles have shed significant insights

into the brain circuits that support monogamous relationships in mammals [27], as well as the effects of parental removal [28], and the physiological and behavioral effects of social isolation and early stress exposures [29]. Although prairie voles are frequently characterized as socially monogamous, mating systems have been reported to vary across populations, particularly among males, with both resident (pair-bonded) and wandering (not pair-bonded) strategies documented in the field [30,31]. Importantly, population-level variation in social structure has not been consistently associated with sexual dimorphism in external morphological traits. Laboratory and field studies comparing prairie voles from Illinois, Tennessee, and Kansas have reported minimal differences in monogamy and parental care [32–36]; however, evidence that these behavioral differences are linked by consistent morphological sexual dimorphism remain equivocal.

In addition to their value in behavioral and neurological research, prairie voles have been the focus of numerous studies on vocal communication. Like many similarly sized rodents, prairie voles produce ultrasonic vocalizations (USVs) at frequencies beyond the range of human's detectable frequency ranges, typically between 25 and 50 kHz [37–40]. In addition, naïve (non-pair-bonded) male and female laboratory prairie voles display the greatest frequency hearing sensitivity between 8–32 kHz and exhibit sex differences in monaural ABR wave characteristics with females generally show larger monaural ABR wave I and II amplitudes and longer wave III and IV latencies than males [14].

Here, we used ABR measurements to compare hearing sensitivity (i.e., ABR thresholds and monaural/binaural ABR amplitudes and latencies) between laboratory-reared and wild-caught prairie voles. We also measured and compared morphological features, including pinna length, pinna width, and craniofacial features between the two groups. Because wild prairie voles are exposed to more selection pressures from their environment, we hypothesized that wild-caught prairie voles will show enhanced auditory sensitivity relative to laboratory-reared prairie voles, characterized by lower ABR thresholds across tested frequencies, faster latencies and higher amplitudes across ABR waves and ITDs, potentially enhancing central and peripheral neural auditory processing. We also hypothesized that there will be no differences in morphological measurements between wild-caught and laboratory prairie voles, due to no sexual dimorphism in external morphological traits documented in this species.

## Materials and methods

The care and use of the animals described in this investigation were reviewed and approved by the Oklahoma State University Institutional Animal Care and Use Committee (IACUC, protocol number 22−09), performed in adherence to the guidelines and recommendations of the American Society of Mammologists (ASM) for the use of wild mammals in research [41], National Institutes of Health (NIH), and the Kansas University Field Research Station. This study was also conducted with permission from the Oklahoma Department of Wildlife Conservation and the Kansas Department of Wildlife and Parks.

### Animal subjects

A total of 53 prairie voles (*Microtus ochrogaster*) were used in this study (Laboratory-reared: N = 33, 16 males and 17 females; wild-caught: N = 20, 10 males and 10 females). Among the laboratory-reared voles, 7 females and 5 males had been mated, while 10 females and 11 males remain non-mated; we included these groups to mimic the diversity in reproductive status likely found in our wild populations. Laboratory-reared prairie voles were obtained from Dr. Tom Curtis in 2020 at the Oklahoma State Health Sciences Campus. These animals originated from long-established laboratory colonies initially derived from wild-caught prairie voles in Illinois in the 1980s and subsequently maintained through outcrossing among multiple university laboratories, with the majority of their genetic background tracing back to wild Illinois populations. Animals were housed in groups of one to three individuals in a temperature-controlled room with a 14:10 hour light/dark cycle and were provided with ad libitum food, water, and nesting material for enrichment. Wild prairie voles were live-trapped between November 2024 to March 2025 at the Kansas University Field Research Station in Lawrence, KS (39.0480534, −95.1930826) using aluminum Sherman (H.B Sherman Traps, Inc. Tallahassee, FL) non-folding traps (3" x

3" x 10"). Traps were baited with a mixture of peanut butter and old-fashioned oats and traps were set for 2 consecutive days for each sampling effort (10 sampling efforts in total). The traps were left overnight and inspected the next morning after around 12 hours. Captured animals were placed in a plastic Ziploc bag for sexing and were then transported to the laboratory for ABR recordings in disposable caging (Animal Specialties and Provisions, LLC, Quakertown, PA, 14.7" L x 9.2" W x 5.5" H). All laboratory-reared prairie voles were adults (> 60 days old) [42]. Among the wild-caught individuals, sixteen (16) were classified as adults and four (4) as subadults. Age classification for wild-caught prairie voles was inferred based on body mass: adults (≥ 30 g) and subadults (21–29 g) [22,43]. Given the unknown mating status, variability and indirect nature of age estimation among wild-caught voles, mating status and age were not included as factors in any data analysis.

**ABR measurement**

Prior to ABR measurements, animals were sedated with a mixture of ketamine (60 mg/kg) and xylazine (10 mg/kg) for initial anesthesia (first 30 minutes) via intraperitoneal injection. Anesthesia was maintained with a follow-up dosage of 25 mg/kg ketamine and 12 mg/kg xylazine administered intraperitoneal for the duration of the experiment (approximately 120 minutes). The depth of anesthesia was assessed at regular intervals using the toe pinch reflex. At the end of experiments, animals were euthanized by overdose of pentobarbital followed by decapitation, consistent with ASM and NIH guidelines. All efforts were made to minimize animal pain and suffering.

ABR testing was used to make comparisons of hearing-related measurements (thresholds, latency, and amplitude responses) between wild-caught and laboratory-reared prairie voles. This noninvasive technique is widely used in both experimental and clinical investigations and involves placing electrodes under the skin to measure electrical activities across various brainstem nuclei in response to sound stimuli. The resulting electrical signals generate ABR waveforms (Fig 1A, 1B), appearing usually in the first 10 ms, where each wave corresponds to synchronized activity and signal transmission from periphery to central ascending auditory areas [44,45].

Details of the ABR procedures have been previously described [46–49]. In brief, the recording was performed using Tucker-Davis Technologies (TDT, Alachua, FL, USA) RA4LI head stage, a RA16PA preamplifier, and Multi I/O processor

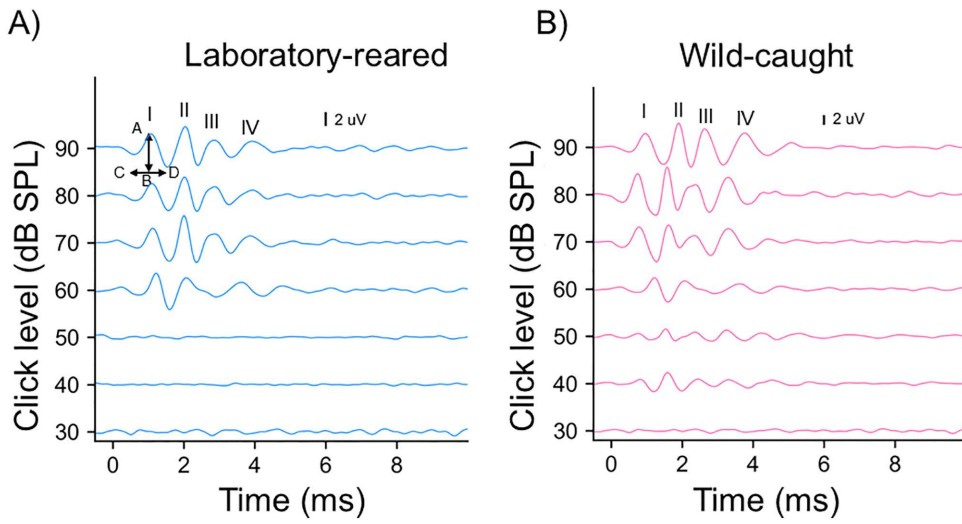

**Fig 1. Figure 1A and 1B demonstrate ABR patterns of a male wild-caught prairie vole (right) and a male laboratory-reared prairie vole (left) measured with clicks of different intensities.** Waves I-IV are labelled in the 90 dB SPL example for each exemplary. The arrow (AB) in Fig 1A shows how amplitude was calculated, while CD arrow shows how latency was calculated.

RZ5 attached to a PC with custom Python software. Animals were placed on a water pump heating pad, which was monitored by a temperature control unit to maintain a body temperature of 37°C. Subdermal needle electrodes (Viasys Healthcare, Madison, WI, USA) were positioned under the skin of the sedated animals for recording, with the active electrodes placed between the pinnae at the vertex of the skull, the reference electrode positioned directly behind the apex of the nape, and the ground electrode inserted in the left back leg of the anesthetized animal. Tucker-Davis Technology Electrostatic Speaker-Coupler Model (TDT MF-1; for frequency 1−24 kHz) and Tucker-Davis Technology Electrostatic Speakers (TDT EC-1; for frequency 32 and 46 kHz) were placed in the left and right ear canals for close-field recording.

Clicks (0.1 ms in duration) and tone bursts (4 ms in duration with a 1 ms on-ramp and 1 ms off-ramp (2 ms ± 1 ms on/off ramps)) stimuli were presented to each animal with alternating polarities. The recorded data were processed using a second-order 50–3000 Hz filter and were averaged across 10–12 ms of recording time over 500–1000 repetitions. Custom earphones were paired with an Etymotic ER-7C probe microphone (Etymotic Research Inc. Elk Grove, IL) for calibration of sound within the ear. Sound stimuli were delivered to the sedated animals with an interstimulus interval of 30 milliseconds and standard deviation of five milliseconds [50]. The stimuli were generated at a sampling rate of 97656.25 Hz using a TDT RP2.1 real-time processor, which was controlled by a custom Python program.

## ABR thresholds

ABR thresholds were determined by visual inspection for each animal by gradually decreasing the stimulus intensity in 10 dB SPL steps, followed by 5 dB SPL steps (when near threshold) to pinpoint the lowest intensity at which wave I of the ABR was detectable in both the left and right monaural stimuli. This process was used to generate frequency response thresholds to compare hearing sensitivity across frequencies (1, 2, 4, 8, 16, 24, 32, 46 kHz) and intensities (10–90 dB SPL) between wild-caught and lab-reared individuals. We next determined the click threshold for each individual by gradually reducing the sound level in 5–10 dB SPL steps until no ABR responses were detected for either ear (Fig 1). Click threshold was estimated as the average between the last level that produced an ABR response and the next lower level that did not. For instance, if a response was observed at 50 dB SPL but not at 40 dB SPL, the threshold was estimated to be 45 dB SPL for this individual.

## Monaural ABR amplitude and latency

To investigate changes in monaural peak amplitude, amplitude ratio, monaural peak latency, and inter-peak latency between wild-caught and lab-reared prairie voles, we recorded monaural click ABR data at 90 dB SPL across the first four peaks of the ABR waveform by presenting the click stimulus independently to each pinna of the anesthetized animal. We quantified the monaural ABR measurement responses for each animal and computed the average of the monaural amplitude and latency of each ABR wave (I-IV) for stimuli delivered to the left and right pinna [14,46,47,51]. Peak amplitudes were determined by measuring the maximum peak-to-trough voltage of each ABR wave, while peak latencies were calculated by measuring the timing of the maximum voltage peak of each ABR wave relative to the timing of the stimulus onset (Fig 1A). Amplitude ratio was calculated by dividing the amplitude of wave I by the amplitude of subsequent peaks (II-IV) for left and right pinnae at 90 dB SPL [52,53]. Inter-peak latency was calculated by computing the difference in latency between peak I and each following peak (II-IV) for the right and left pinnae at 90 dB SPL [4,54,55].

## Binaural ABR amplitude and latency

We presented broadband click stimuli at 90 dB SPL to both pinnae simultaneously with an interaural time difference (ITD) ranging between −2.0 to +2.0 ms in 0.5 ms increments to evoke binaural ABR responses. The BIC was calculated by deducting the sum of the two monaural responses from the binaural ABR recordings similar to previous publications [46,47,50–51,56]. The BIC was characterized as the prominent negative peak (DN1) occurring at wave IV of the binaural ABR after subtracting the sum of the monaural and binaural responses [50,56]. BIC amplitude and latency were quantified

across ITDs, with BIC amplitude determined as the peak relative to 0 ITD and the baseline of the overall trace, which was set to 0. The average DN1 normalized latency and relative DN1 amplitude values were used to compare binaural ABR responses as a function of ITD between laboratory-reared and wild-caught prairie voles.

### Craniofacial and pinnae measurements

We obtained craniofacial and pinnae measurements for each animal to test for morphological differences between wild-caught and laboratory-reared animals using a six-inch Stainless Steel Electronic Vernier Caliper (DIGI-Science Accumatic digital Caliper Gyros Precision Tools Monsey, New York, USA). Measurements included pinna length (distance between basal notch to the tip without hair), pinna width (distance between the subhelix and the tip excluding hair), effective pinna diameter (square root of pinna width x pinna length), inter-pinna distance (distance between the two ear canals), nose to pinna distance (distance from the tip of the nose to the midpoint between the pinnae) [57–59]. Lastly, body mass was measured using a digital stainless steel electronic scale (Weighmax W-2809 90 LB X 0.1 OZ Durable Stainless Steel Digital Postal scale, Chino, California, USA) for each animal.

### Data analyses

Comparisons of threshold levels for the tested frequencies, monaural and binaural amplitudes and latencies, as well as morphological features were performed in R [60]. We compared ABR response thresholds between wild-caught and laboratory-reared prairie voles using linear mixed-effect models (LMMs, lmerTest package), where threshold responses served as the response variable, frequency, sex, and origin (laboratory or wild voles) were used as fixed effects, and individual and side (right and left monaural ABR) were treated as random effects [61]. Body mass was not included in any LMM models due to no difference between groups and no improvement of the best model fit. Similarly, we analyzed differences in monaural and binaural ABR measures (amplitude and latency) using linear mixed-effect models, incorporating sex, origin, ITD, and peak as fixed effects, and individual as a random effect. Fixed-effect structures were evaluated using likelihood ratio tests to compare nested models, and all post hoc analyses were performed on the minimal adequate model. Estimated marginal means were obtained using the emmeans package [62] and were used to perform pairwise comparisons across frequencies, monaural ABR peaks, ITDs between treatment group and sex when there was a significant interaction between fixed effects. To account for multiple comparisons, the emmeans package applied Tukey's Honestly Significant Difference (HSD) method for contrasts. Shapiro-Wilk test was used to test for normality of the click thresholds, morphological measurements features, and body mass data. Because these variables were not normally distributed, group comparisons were conducted using Mann Whitney U test, Assumptions of linear mixed-effects models were evaluated using residual diagnostic plots (Q-Q plots), which indicated no major violations. All figures were created in Python and RStudio using the ggplot2 package [63].

## Results

### Hearing thresholds

Frequency response thresholds were determined from ABR waveforms in both laboratory-reared and wild-caught prairie voles. Both groups exhibited the greatest sensitivity to tone stimuli at mid-range frequencies with the lowest ABR detection thresholds between 8–24 kHz in wild-caught and 8–32 kHz in laboratory-reared prairie voles (Fig 2A). Linear mixed effect models detected significant main effects of frequency (LMM: N = 53, $\chi^2$ (7) = 256.26, p < 0.001), origin (LMM: N = 53, $\chi^2$ (1) = 7.008, p = 0.008), and the interaction of origin and frequency (LMM: N = 53, $\chi^2$ (7) = 38.72, p < 0.001) on ABR response thresholds between laboratory-reared and wild-caught prairie voles. Pairwise comparisons indicated that wild-caught *M. ochrogaster* displaying significantly lower ABR thresholds at 1, 4, 8, 16, and 24 kHz, but not at 32 kHz than laboratory-reared voles (Table 1). There were no significant differences in ABR thresholds at 2 and 46 kHz between groups.

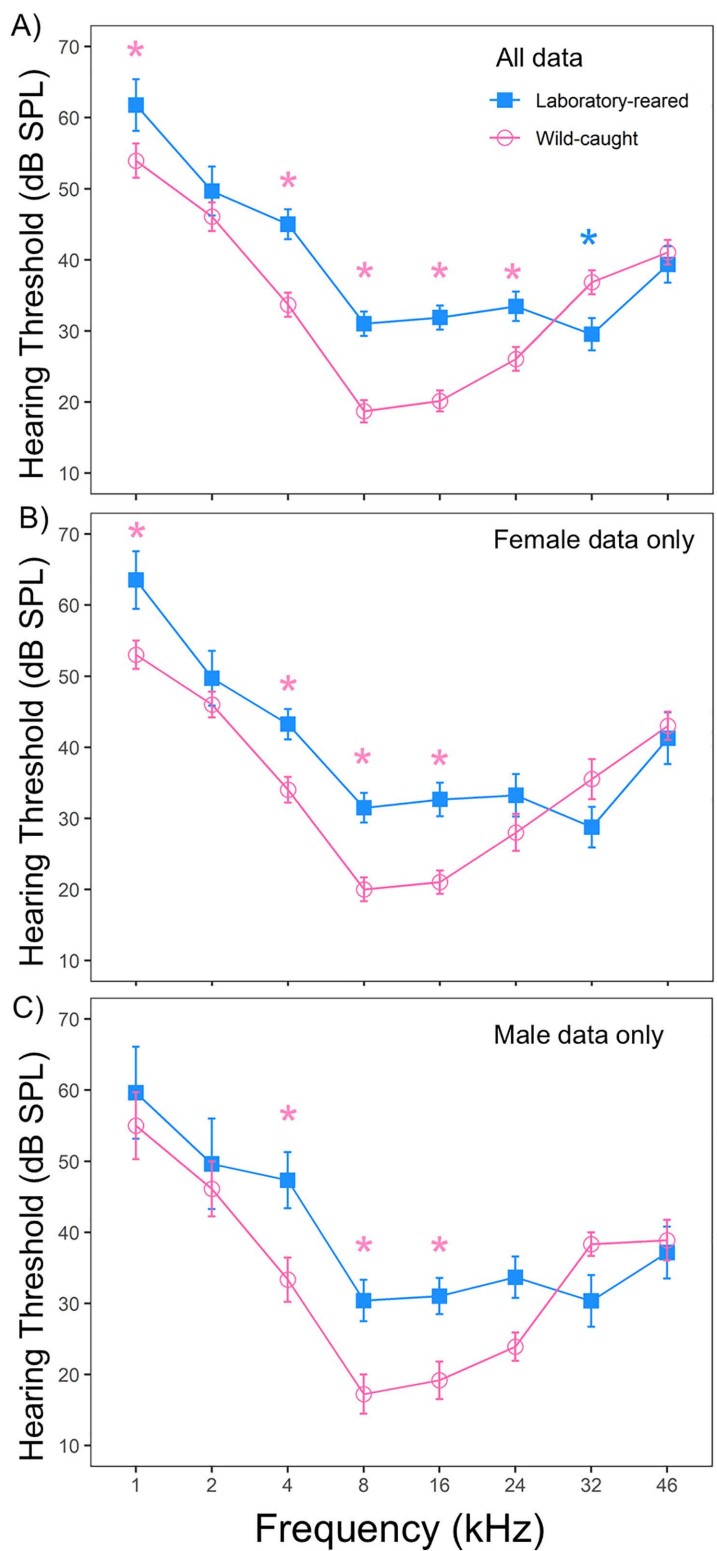

**Fig 2. Auditory brainstem response thresholds in laboratory-reared and wild-caught prairie voles (_M. ochrogaster_). (A)** Mean ABR thresholds across frequencies for all laboratory-reared (N = 33) and wild-caught (N = 20) prairie voles. **(B)** Frequency-specific ABR thresholds for female prairie voles by origin (laboratory-reared N = 17, wild-caught N = 10). **(C)** Frequency-specific ABR thresholds for male prairie voles by origin (laboratory-reared N = 16,

wild-caught N = 10). Error bars represent the standard error of the mean (SEM) at each tested frequency. Significant main effects of frequency and origin were detected, with wild-caught voles generally exhibiting lower thresholds than laboratory-reared counterparts across most frequencies. Asterisks indicate statistically significant differences in mean ABR thresholds between groups (p < 0.05).

Table 1. Statistical measures of auditory brainstem response thresholds between laboratory-reared (N = 33) and wild-caught (N = 20) prairie voles. Values displayed represent the mean at each frequency tested ± standard error, the t-ratio value, and the p-value of ABR response thresholds between lab-reared and wild-caught prairie voles.

| Frequency (kHz) | Laboratory-reared mean ± S. E | Wild-caught mean ± S. E | t-ratio | p-value |
|---|---|---|---|---|
| 1 | 61.8 ± 2.16 | 53.9 ± 2.77 | 2.230 | 0.0266 |
| 2 | 49.8 ± 2.19 | 46.1 ± 2.79 | 1.070 | 0.2854 |
| 4 | 45.2 ± 2.19 | 33.7 ± 2.79 | 3.242 | 0.0013 |
| 8 | 31.4 ± 2.19 | 18.7 ± 2.79 | 3.593 | 0.0004 |
| 16 | 32.0 ± 2.13 | 20.1 ± 2.79 | 3.381 | 0.0008 |
| 24 | 33.6 ± 2.13 | 26.1 ± 2.79 | 2.140 | 0.0333 |
| 32 | 29.9 ± 2.16 | 36.8 ± 2.78 | −1.978 | 0.0490 |
| 46 | 39.6 ± 2.19 | 41.1 ± 2.79 | −0.419 | 0.6755 |

ABR response thresholds between the sexes were tested between laboratory-reared and wild-caught prairie voles. The results of the linear mixed-effects model revealed no significant statistical differences in ABR threshold between the sexes (LMM: N = 53, $\chi^2$ (1) = 0.12, p = 0.725). There was a significant main effect of frequency (LMM: N = 53, $\chi^2$ (7) = 263.93, p < 0.001) on ABR threshold between sexes, but no main interaction effect of frequency and sex (LMM: N = 53, $\chi^2$ (7) = 3.51, p = 0.834). Because there was no interaction effect, pairwise comparisons were not performed between sexes. When comparing within females, we detected significant main effects of frequency (LMM: N = 27, $\chi^2$ (7) = 155.16, p < 0.001), origin (LMM: N = 27, $\chi^2$ (1) = 5.41, p = 0.019), and the interaction of origin and frequency (LMM: N = 27, $\chi^2$ (7) = 20.19, p = 0.005) on ABR thresholds across most tested frequencies. Female wild-caught voles displayed significantly lower ABR thresholds than female lab-reared prairie voles at 1, 4, 8, and 16 kHz (all p < 0.05, Fig 2B, supplementary S1 Table in S1 File). Similarly, we observed significant main effects of frequency (LMM: N = 26, $\chi^2$ (7) = 114.64, p < 0.001) and the interaction of frequency and origin (LMM: N = 26, $\chi^2$ (7) = 21.71, p = 0.002) on ABR thresholds between male wild-caught and male lab-reared voles; however, no significant main effect of origin was detected between male lab-reared and male wild-caught prairie voles (LMM: N = 27, $\chi^2$ (1) = 2.64, p = 0.104). Male wild-caught voles displayed significantly lower ABR thresholds than male lab-reared prairie voles at 4, 8, and 16 kHz (all p < 0.05, Fig 2C, supplementary Table S2 in S1 File).

### Click thresholds

We found significant statistical differences in the click thresholds between the two tested groups (Mann Whitney U: N = 50, W = 542, p < 0.001). On average, wild-caught prairie voles exhibited a click threshold of 52 dB SPL, with a click threshold of 67 dB SPL for laboratory-reared prairie voles (Fig 3A). There were no sex differences in the click threshold between laboratory-reared and wild-caught *M. ochrogaster* (Mann Whitney U: N = 50, W = 375.5, p = 0.345). However, within-sex comparisons revealed significant differences in overall click thresholds between laboratory-reared and wild-caught prairie voles in both females (Fig 3B: Mann Whitney U: N = 27, W = 155, p < 0.001) and males (Fig 3C: Mann Whitney U: N = 23, W = 114, p = 0.001).

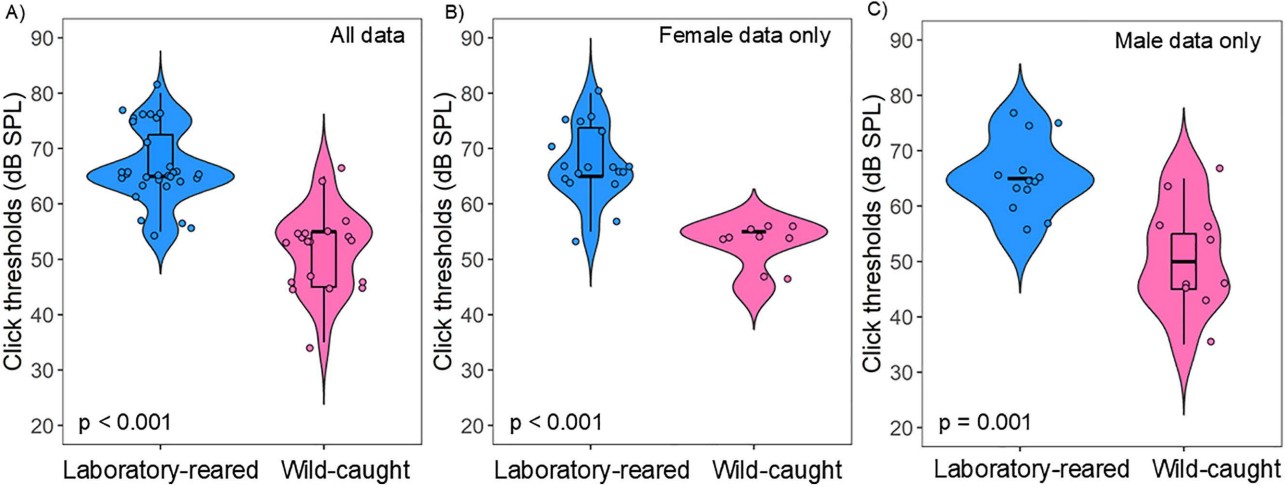

**Fig 3. Click response thresholds in laboratory-reared and wild-caught prairie voles (*M. ochrogaster*). (A)** Mean click thresholds for all laboratory-reared (N = 30) and wild-caught (N = 20) prairie voles. **(B)** Click thresholds for female prairie voles by origin (laboratory-reared N = 17, wild-caught N = 10). (C) Click thresholds for male prairie voles by origin (laboratory-reared N = 13, wild-caught N = 10). Significant differences in mean click thresholds were detected, with wild-caught voles generally exhibiting lower click thresholds than laboratory-reared counterparts.

## Monaural ABR latency

Individual features of ABR waveforms (wave I-IV), evoked by 90 dB SPL click, were assessed. ABR wave latencies, which reflect how quickly signals are transmitted along the ascending auditory pathway, were measured between wild caught and lab-reared animals. Linear mixed-effects models showed significant main effects of peak number (LMM: N = 53, $\chi^2$ (3) = 1108.8, p < 0.001) and the interaction of origin and peak number (LMM: N = 53, $\chi^2$ (3) = 16.41, p < 0.001) in monaural ABR absolute latency responses. However, there was no main effect of origin alone on monaural ABR absolute latencies (LMM: N = 53, $\chi^2$ (1) = 1.94, p = 0.164). Wild-caught prairie voles showed longer monaural absolute latency at ABR wave IV than laboratory-reared prairie voles (t-value = −2.499, p = 0.014; Fig 4A), however, no statistical differences were observed in monaural latency of ABR wave I (t-value = 0.663, p = 0.509), wave II (t-value = −1.424, p = 0.158), and wave III (t-value = −1.538, p = 0.127). There were no significant main effects of sex (LMM: N = 53, $\chi^2$ (1) = 0.165, p = 0.685) and the interaction of sex and peak number (LMM: N = 53, $\chi^2$ (3) = 11.98, p = 0.576) in absolute latency between groups. However, there was a main effect of peak number alone on monaural ABR absolute latencies (LMM: N = 53, $\chi^2$ (3) = 1107.7, p < 0.001) between the sexes. Because there was no main interaction effect, we did not perform pairwise comparisons between the sexes. Within female comparisons showed no significant main effects of origin (LMM: N = 27, $\chi^2$ (1) = 0.222, p = 0.638, Fig 4B), and the interaction of origin and peak number in absolute latency (LMM: N = 27, $\chi^2$ (3) = 1.781, p = 0.619). There was a main effect of peak number alone on monaural ABR absolute latencies (LMM: N = 27, $\chi^2$ (3) = 505.76, p < 0.001) within female prairie voles. We did not perform pairwise comparisons within female prairie voles since there was no main interaction effect. Linear-mixed effect models revealed significant main effects of origin (LMM: N = 26, $\chi^2$ (1) = 9.748, p = 0.001), peak number (LMM: N = 26, $\chi^2$ (3) = 653.66, p < 0.001), and the interaction of origin and peak number (LMM: N = 26, $\chi^2$ (3) = 56.05, p < 0.001) in absolute latency between male laboratory-reared and male wild-caught prairie voles. Wild-caught males exhibited longer absolute latency at wave II (t-value = −2.499, p = 0.017), wave III (t-value = −3.629, p < 0.001), and wave IV (t-value = −5.683, p < 0.001) compared to laboratory males prairie voles (Fig 4C).

To compare latency time responses between both tested groups, interpeak latencies were calculated by subtracting the latency of peak I and each subsequent latency peak (II-IV) for at 90 dB SPL. Significant main effects of origin (LMM: N = 49, $\chi^2$ (1) = 10.17, p = 0.001), peak number (LMM: N = 49, $\chi^2$ (2) = 720.36, p < 0.001), and the interaction of origin and

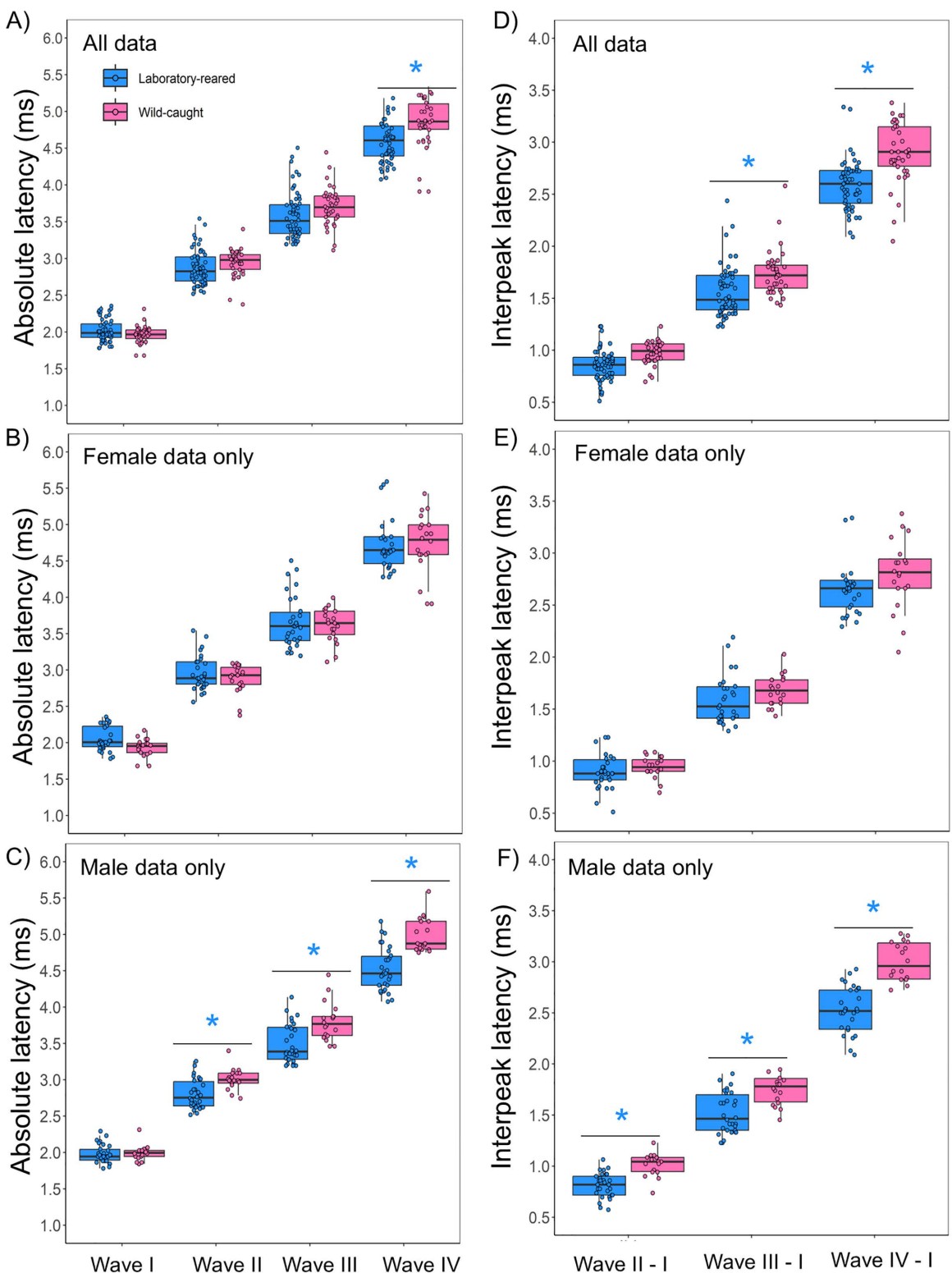

**Fig 4. Absolute and interpeak latencies in laboratory-reared and wild-caught prairie voles (*M. ochrogaster*). (A)** Mean absolute latency across peak numbers for all laboratory-reared (N = 33) and wild-caught (N = 20) prairie voles. **(B)** Absolute latency responses for female prairie voles by origin (laboratory-reared N = 17, wild-caught n = 10). **(C)** Absolute latency responses for male prairie voles by origin (laboratory-reared N = 16, wild-caught

N = 10). **(D)** Mean interpeak latencies across peak numbers for all laboratory-reared (N = 29) and wild-caught (N = 20) prairie voles. **(E)** Interpeak latency responses for female prairie voles by origin (laboratory-reared N = 15, wild-caught N = 10). **(F)** Interpeak latency responses for male prairie voles by origin (laboratory-reared N = 14, wild-caught N = 10). Significant main effects of peak number and origin were detected, with wild-caught voles generally exhibiting longer absolute and interpeak latency responses than laboratory-reared counterparts across most ABR peaks. Asterisks indicate statistically significant differences in mean monaural ABR absolute and interpeak latencies between groups (p < 0.05).

peak number (LMM: N = 49, $\chi^2$ (2) = 9.83, p = 0.007) were detected in interpeak latency between laboratory-reared and wild-caught *M. ochrogaster*. Laboratory-reared prairie voles showed faster interpeak latency (Fig 4D) at peak III – I (mean difference = −0.177, t-value = −2.606, p = 0.011), and peak IV – I (mean difference = −0.294, t-value = −4.316, p < 0.001). There were no main effects of sex (LMM: N = 49, $\chi^2$ (1) = 0.845, p = 0.358) and the interaction of sex and peak number (LMM: N = 49, $\chi^2$ (2) = 2.22, p = 0.329) in interpeak latencies between the tested groups. However, there was a main effect of peak number alone on interpeak latency between the sexes (LMM: N = 49, $\chi^2$ (2) = 714.88, p < 0.001). As there was no main interaction effect, we did not perform pairwise comparisons between sexes. Within-female comparisons revealed no significant main effects of origin (LMM: N = 25, $\chi^2$ (1) = 0.323, p = 0.657) and the interaction of origin and peak number (LMM: N = 25, $\chi^2$ (2) = 0.557, p = 0.757) on interpeak latencies (Fig 4E). There was a significant peak number effect on interpeak latency within female prairie voles (LMM: N = 25, $\chi^2$ (2) = 368.19, p < 0.001). Because there was no main interaction effect, we did not perform pairwise comparisons. When comparing male lab-reared to male wild-caught prairie voles, the linear mixed effect models showed significant main effects of origin (LMM: N = 24, $\chi^2$ (1) = 18.63, p < 0.001), peak number (LMM: N = 24, $\chi^2$ (2) = 361.34, p < 0.001), and the interaction of origin and peak number (LMM: N = 24, $\chi^2$ (2) = 17.87, p < 0.001) on the interpeak latencies. Laboratory-reared male prairie voles showed faster interpeak latency at wave II – I (mean difference = −0.219, t-value = −2.663; p = 0.010), wave III – I (mean difference = −0.29, t-value = −3.519, p < 0.001), and wave IV – I (mean difference = −0.539, t-value = −6.563, p < 0.001) (Fig 4F).

## Monaural ABR amplitude

When comparing absolute monaural amplitudes between groups, the results of the linear mixed effects models revealed significant main effects of origin (LMM: N = 49, $\chi^2$ (1) = 19.62, p < 0.001), peak number (LMM: N = 49, $\chi^2$ (3) = 67.91, p < 0.001), and the interaction of origin and peak number (LMM: N = 49, $\chi^2$ (3) = 18.57, p < 0.001). Wild-caught prairie voles displayed a larger monaural ABR peak amplitude compared to laboratory-reared voles for wave I (t-value = −3.664, p < 0.001), wave II (t-value = − 6.166, p < 0.001), and wave III (t-value = −3.162, p = 0.001) (Fig 5A). However, no difference was detected in ABR wave amplitude IV between tested groups (t-value = −1.666, p = 0.098). There were no significant main effects of sex (LMM: N = 49, $\chi^2$ (1) = 1.048, p = 0.306) or the interaction of sex and peak number (LMM: N = 49, $\chi^2$ (3) = 3.865, p = 0.276) in absolute ABR wave amplitudes. However, there was a main effect of peak number alone in absolute ABR wave amplitude between the sexes (LMM: N = 49, $\chi^2$ (3) = 67.914, p < 0.001). Pairwise comparisons were not performed as there was no interaction effect between the sexes on ABR absolute amplitude. For female prairie voles, linear-mixed effect models revealed significant main effects of origin (LMM: N = 25, $\chi^2$ (1) = 8.199, p = 0.004), and peak number (LMM: N = 25, $\chi^2$ (3) = 48.35, p < 0.001, Fig 5B). However, there was no interaction effect of origin and peak number (LMM: N = 25, $\chi^2$ (3) = 5.857, p = 0.119) in absolute amplitude between female laboratory-reared and female wild-caught prairie voles. Because there was no interaction effect of peak number and origin within female prairie voles, we did not perform pairwise comparisons. We detected significant main effects of origin (LMM: N = 24, $\chi^2$ (1) = 11.879, p < 0.001), peak number (LMM: N = 24, $\chi^2$ (3) = 23.16, p < 0.001), and the interaction of origin and peak number (LMM: N = 24, $\chi^2$ (3) = 13.89, p = 0.003) when comparing male wild-caught and laboratory-reared voles. Wild-caught males exhibited significantly larger amplitudes than their laboratory counterparts for wave I (t-value = −2.895, p = 0.005), wave II (t-value = −5.051, p < 0.001), and wave III (t-value = −2.425, p = 0.018) (Fig 5C). However, no difference was observed for monaural ABR wave IV absolute amplitude between groups (t-value = −1.333, p = 0.187).

 

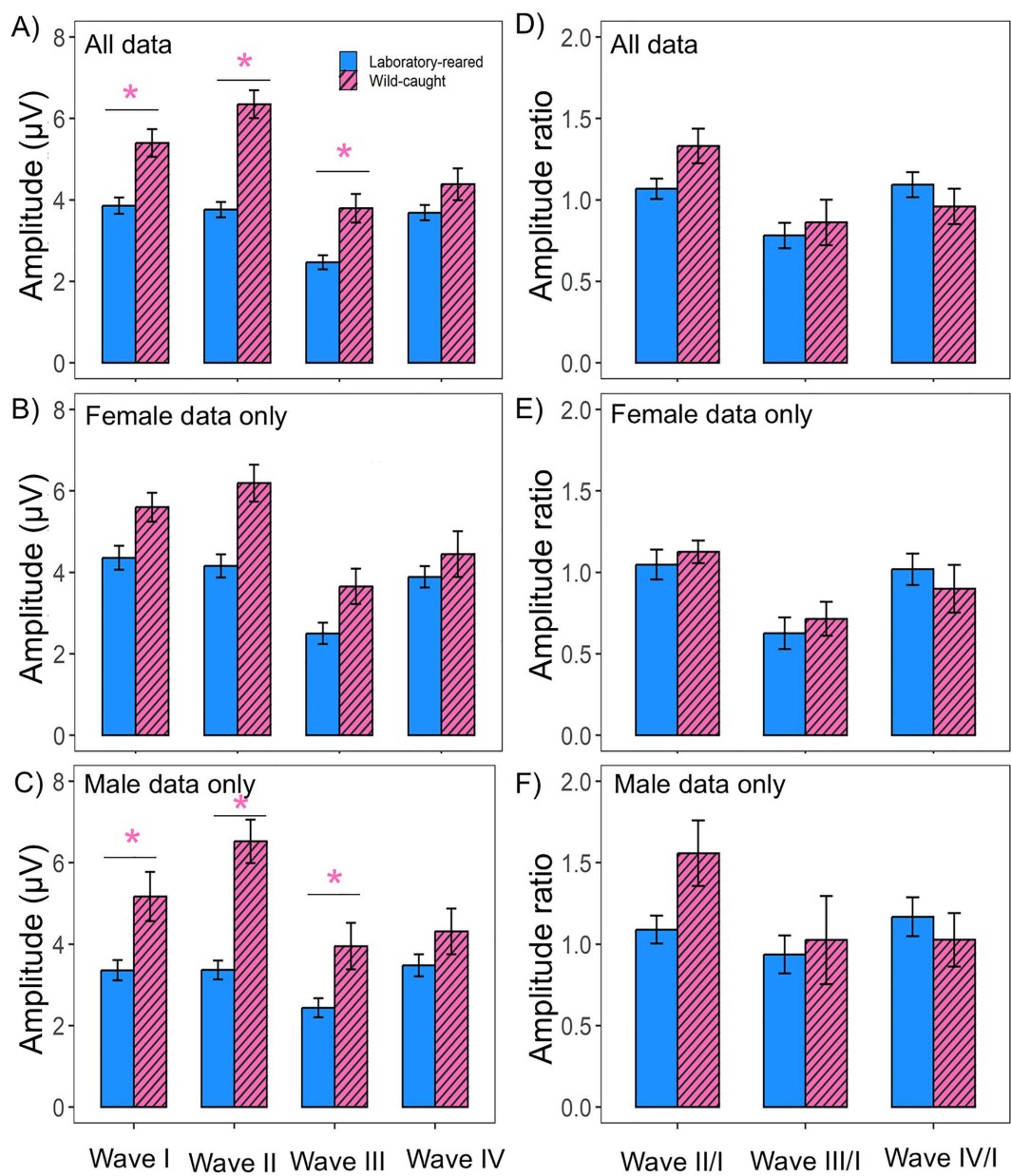

**Fig 5. Absolute amplitude and amplitude ratio in laboratory-reared and wild-caught prairie voles (*M. ochrogaster*). (A)** Mean absolute amplitude across peak numbers for all laboratory-reared (N = 29) and wild-caught (N = 20) prairie voles. **(B)** Absolute amplitude responses for female prairie voles by origin (laboratory-reared N = 15, wild-caught N = 10). **(C)** Absolute amplitude responses for male prairie voles by origin (laboratory-reared N = 14, wild-caught N = 10). **(D)** Mean amplitude ratio across peak numbers for all laboratory-reared (N = 29) and wild-caught (N = 20) prairie voles. **(E)** Amplitude ratio responses for female prairie voles by origin (laboratory-reared N = 15, wild-caught N = 10). **(F)** Amplitude ratio responses for male prairie voles by origin (laboratory-reared N = 14, wild-caught N = 10). Significant main effects of peak number and origin were detected, with wild-caught voles generally exhibiting higher absolute and amplitude ratio responses than laboratory-reared counterparts across most ABR peaks. Asterisks indicate statistically significant differences in mean monaural ABR absolute and interpeak latencies between groups (All p < 0.05).

ABR wave amplitude ratios were calculated to compare amplitude responses between groups. There was no significant main effect of origin on amplitude ratio between laboratory-reared and wild-caught prairie voles (LMM: N = 49, $\chi^2$ (1) = 0.231, p = 0.631). However, we saw main effects of peak number (LMM: N = 49, $\chi^2$ (2) = 31.65, p < 0.001), and the interaction of peak number and origin (LMM: N = 49, $\chi^2$ (2) = 9.767, p = 0.007) on amplitude ratio between groups (Fig 5D). Because there was a main effect of origin and peak number, we performed pairwise comparisons on the amplitude ratio data across peaks, however, we did not find any statistical differences in pairwise comparisons amplitude ratio across waves (LMM: p-values > 0.05). There were no main effects of sex (LMM: N = 49, $\chi^2$ (1) = 2.32, p = 0.128) or the interaction of sex and peak number (LMM: N = 49, $\chi^2$ (2) = 1.88, p = 0.389) on the amplitude wave ratios between groups. However, there was a main effect of peak number alone on amplitude ratio between the sexes (LMM: N = 49, $\chi^2$ (2) = 31.65, p < 0.001). Pairwise comparisons on amplitude ratio between sexes were not performed since there was no significant interaction between sex and peak number. Comparisons between female laboratory-reared and female wild-caught revealed no significant main effects of origin (LMM: N = 25, $\chi^2$ (1) = 0.011, p = 0.918) or the interaction effect (LMM: N = 25, $\chi^2$ (2) = 2.365, p = 0.306) between origin and peak number in amplitude ratio across ABR waves (Fig 5E). However, a significant effect of peak number alone was observed on amplitude ratio between female voles (LMM: N = 25, $\chi^2$ (2) = 29.76, p < 0.001). Pairwise comparisons on amplitude ratio between female prairie voles were not performed since there was no significant interaction between origin and peak number. Linear mixed-effect models revealed significant main effect of peak number (LMM: N = 24, $\chi^2$ (2) = 8.418, p = 0.015) and the interaction of origin and peak number (LMM: N = 24, $\chi^2$ (2) = 9.226, p = 0.009), but no main effect of origin (LMM: N = 24, $\chi^2$ (1) = 0.329, p = 0.566) on amplitude ratio between male laboratory-reared and wild-caught prairie voles (Fig 5F). With a significant interaction between peak number and origin, we performed pairwise comparisons across wave ratios, however, we did not find any statistical differences for any ABR wave ratio tested between male laboratory-reared and male wild-caught voles (All p > 0.05).

## Binaural ABR normalized DN1 latency

To explore the relationship of ITD on binaural hearing between laboratory-reared and wild-caught prairie voles, we examined changes in normalized latency and relative amplitude of DN1 components of the BIC. As expected, the shift in normalized latency exhibited a positive relationship with increasing ITD, while relative amplitude displayed a negative relationship with longer ITD across groups. We found significant main effects of ITD (LMM: N = 45, $\chi^2$ (4) = 358.1, p < 0.001), origin (LMM: N = 45, $\chi^2$ (1) = 7.312, p = 0.007), and the interaction of origin and ITD (LMM: N = 45, $\chi^2$ (4) = 37.56, p < 0.001) on normalized latency between groups, suggesting longer latency as ITD increased. Pairwise comparisons revealed that the shift in normalized DN1 latency was significantly faster in laboratory-reared prairie voles compared to wild-caught at 1.5 ms (t-ratio = −2.714, p = 0.007) and 2.0 ms (t-ratio = −6.198, p < 0.001) ITDs (Fig 6A). There were no significant main effects of sex (LMM: N = 45, $\chi^2$ (1) = 0.035, p = 0.851), or the interaction of sex and ITD (LMM: N = 45, $\chi^2$ (4) = 0.597, p = 0.963) on latency changes of the DN1 component with increasing ITD between groups. However, there was a main effect of ITD alone on normalized DN1 latency between the sexes (LMM: N = 45, $\chi^2$ (4) = 353.1, p < 0.001). Pairwise comparisons on normalized DN1 latency between the sexes were not performed since there was no significant interaction between sex and ITD. Within-female comparisons revealed significant main effects of ITD (LMM: N = 23, $\chi^2$ (4) = 180.12, p < 0.001), and the interaction of ITD and origin (LMM: N = 23, $\chi^2$ (4) = 18.07, p = 0.001), but not origin (LMM: N = 23, $\chi^2$ (1) = 2.80, p = 0.094) on DN1 latency shifts. Female laboratory-reared voles showed faster DN1 latency at 2.0 ms ITD (t-value = −4.077, p < 0.001) compared to female wild-caught voles, however, both groups exhibited similar DN1 latency at other ITDs (All p > 0.05) (Fig 6B). Likewise, there were significant main effects of ITD (LMM: N = 22, $\chi^2$ (4) = 188.56, p < 0.001), origin (LMM: N = 22, $\chi^2$ (1) = 4.681, p = 0.030) and the interaction of ITD and origin (LMM: N = 22, $\chi^2$ (4) = 22.06, p < 0.001) on DN1 latency between male laboratory-reared and male wild-caught prairie voles. Male laboratory reared prairie voles showed faster DN1 latency at 1.5 ms (t-value = −2.169, p = 0.034) and 2.0 ms (t-value = − 4.544, p < 0.001) ITDs, however, both groups displayed similar DN1 latency at other ITDs (0 ms, 0.5 ms, and 1.0 ms; All p > 0.05) (Fig 6C).

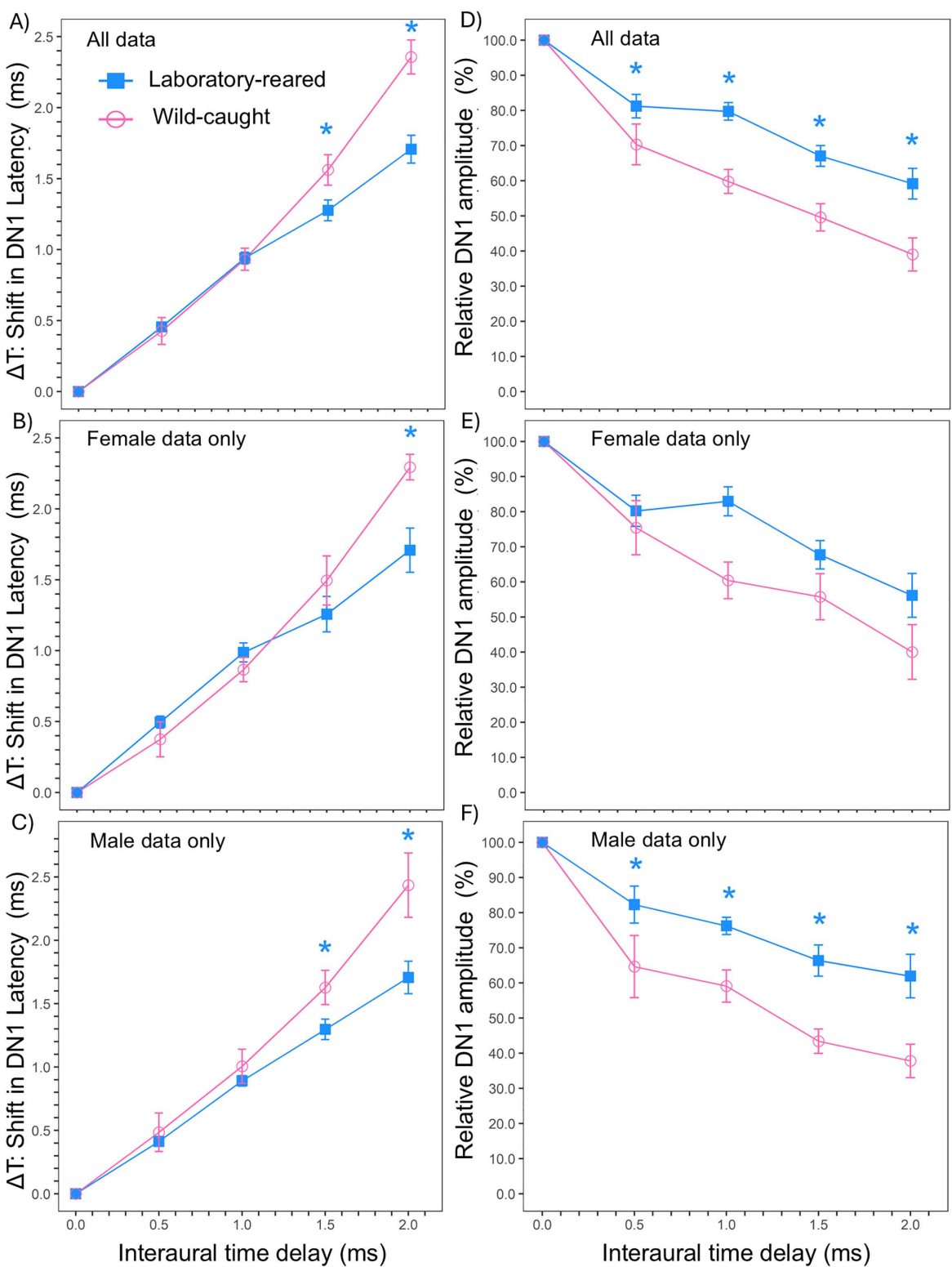

**Fig 6. Binaural ABR latency and relative amplitude between lab-reared and wild-caught prairie voles (*M. ochrogaster*). (A)** The change in DN1 latency (ms) in relation to ITD; reference latency at ITD = 0 is set to 0 ms with varying ITD (−2.0 to + 2.0) for all laboratory-reared (N = 25) and wild-caught (N = 20) prairie voles. **(B)** DN1 latency for female prairie voles by origin (laboratory-reared N = 13, wild-caught N = 10). **(C)** DN1 latency for male prairie

voles by origin (laboratory-reared n = 12, wild-caught N = 10). **(D)** The change in relative DN1 amplitude (%) in relation to ITD; reference latency at ITD = 0 is set to 0 ms with varying ITD (−2.0 to + 2.0) for all laboratory-reared (N = 25) and wild-caught (N = 20) prairie voles. **(E)** Relative amplitude for female prairie voles by origin (laboratory-reared n = 13, wild-caught N = 10). **(F)** Relative amplitude for male prairie voles by origin (laboratory-reared n = 12, wild-caught N = 10). Significant main effects of ITD and Origin were detected in BIC shift DN1 latencies and origin between the groups. No main effect of sex was observed between groups for either DN1 latency or relative amplitude across ITD. Asterisks indicate statistically significant differences in mean monaural ABR absolute and interpeak latencies between groups (p < 0.05).

## Binaural ABR relative DN1 amplitude

The results of the linear-mixed effect models showed significant main effects of ITD (LMM: N = 49, $\chi^2$ (4) = 166.05, p < 0.001), origin (LMM: N = 49, $\chi^2$ (1) = 20.92, p < 0.001), and the interaction of origin and ITD (LMM: N = 45, $\chi^2$ (4) = 14.38, p = 0.006) on DN1 relative amplitude between groups. Generally, relative amplitude decreased with longer ITDs, as demonstrated in Fig 6 (D, E, F). Relative DN1 amplitude of th e BIC was significantly lower in wild-caught com-pared to laboratory-reared individuals at 0.5 ms (t-value = 2.182, p = 0.030), 1.0 ms (t-value = 4.116, p < 0.001), 1.5 ms (t-value = 3.524, p < 0.001), and 2.0 ms (t-value = 4.101, p < 0.001) ITD (Fig 6D). There were no main effect of sex (LMM: N = 45, $\chi^2$ (1) = 0.357, p = 0.055) or the interaction of sex and ITDs (LMM: N = 45, $\chi^2$ (4) = 2.479, p = 0.648) on relative DN1 amplitude between groups, but we observed significant main effects of ITD (LMM: N = 45, $\chi^2$ (4) = 162.01, p < 0.001) on relative amplitude. As we found no interaction effect of sex and ITD on relative amplitude, pairwise comparisons were not made. When comparing within females, significant main effects of ITD (LMM: N = 23, $\chi^2$ (4) = 81.08, p < 0.001) and origin (LMM: N = 23, $\chi^2$ (1) = 7.739, p = 0.005), but not the interaction of ITD and origin (LMM: N = 23, $\chi^2$ (4) = 7.276, p = 0.122) were observed for relative DN1 amplitude (Fig 6E). Pairwise comparisons were not performed since there was no inter-action effect of origin and ITDs on relative amplitude within females. For males, there were significant main effects of ITD (LMM: N = 22, $\chi^2$ (4) = 89.11, p < 0.001), origin (LMM: N = 22, $\chi^2$ (1) = 13.97, p < 0.001), and the interaction of ITD and origin (LMM: N = 22, $\chi^2$ (4) = 10.672, p = 0.030) in relative DN1 amplitude. Male laboratory-reared voles showed higher relative amplitude than male wild-caught voles at 0.5 ms (t-value = 2.531; p = 0.013), 1.0 ms (t-value = 2.607, p = 0.010), 1.5 ms (t-value = 3.453, p < 0.001), and 2.0 ms (t-value = 3.590, p < 0.001) ITDs, but not 0 ms (p > 0.05) (Fig 6F).

## Morphology features

There were significant differences in pinna width (Mann Whitney U: N = 53, W = 535.5, p = 0.020, Fig 7B) and inter-pinna distance (Mann Whitney U: N = 53, W = 241, p = 0.017, Fig 7D) between laboratory-reared and wild-caught prairie voles. In general, laboratory-reared prairie voles exhibit larger pinna and shorter inter-pinna distance than wild-caught voles. We found no statistically significant differences for pinna length (Mann Whitney U: N = 53, W = 382.5, p = 0.911 Fig 7A), effective pinna diameter (Mann Whitney U: N = 53, W = 495, p = 0.094, Fig 7C), nose-to-pinna distance (Mann Whitney U: N = 53, W = 277.5, p = 0.073, Fig 7E) or body mass (Mann Whitney U: N = 53, W = 371, p = 0.766, Fig 7F) between laboratory-reared and wild-caught prairie voles. No sex differences were observed in any of the craniofacial character-istics, pinna features, or body mass between prairie voles reared in the laboratory and caught in the wild (All p > 0.05). Within-sex comparisons revealed no significant differences in morphological feature measurements or body mass between laboratory-reared and wild-caught prairie voles in both females (All p > 0.05, except nose to pinna distance; Mann Whitney U: N = 23, W = 52.5, p = 0.038) and males (All p > 0.05).

## Discussion

Our results showed main effects of frequency, origin, and the interaction between origin and frequency on pure tone thresholds between groups. Wild-caught prairie voles typically exhibited lower frequency hearing thresholds compared to lab-reared voles at the lower and higher frequency ranges tested (except 32 kHz), with significant differences observed at 1, 4, 8, 16, 24 and 32 kHz. To our knowledge, this is the first reported evidence of ABR-frequency derived hearing

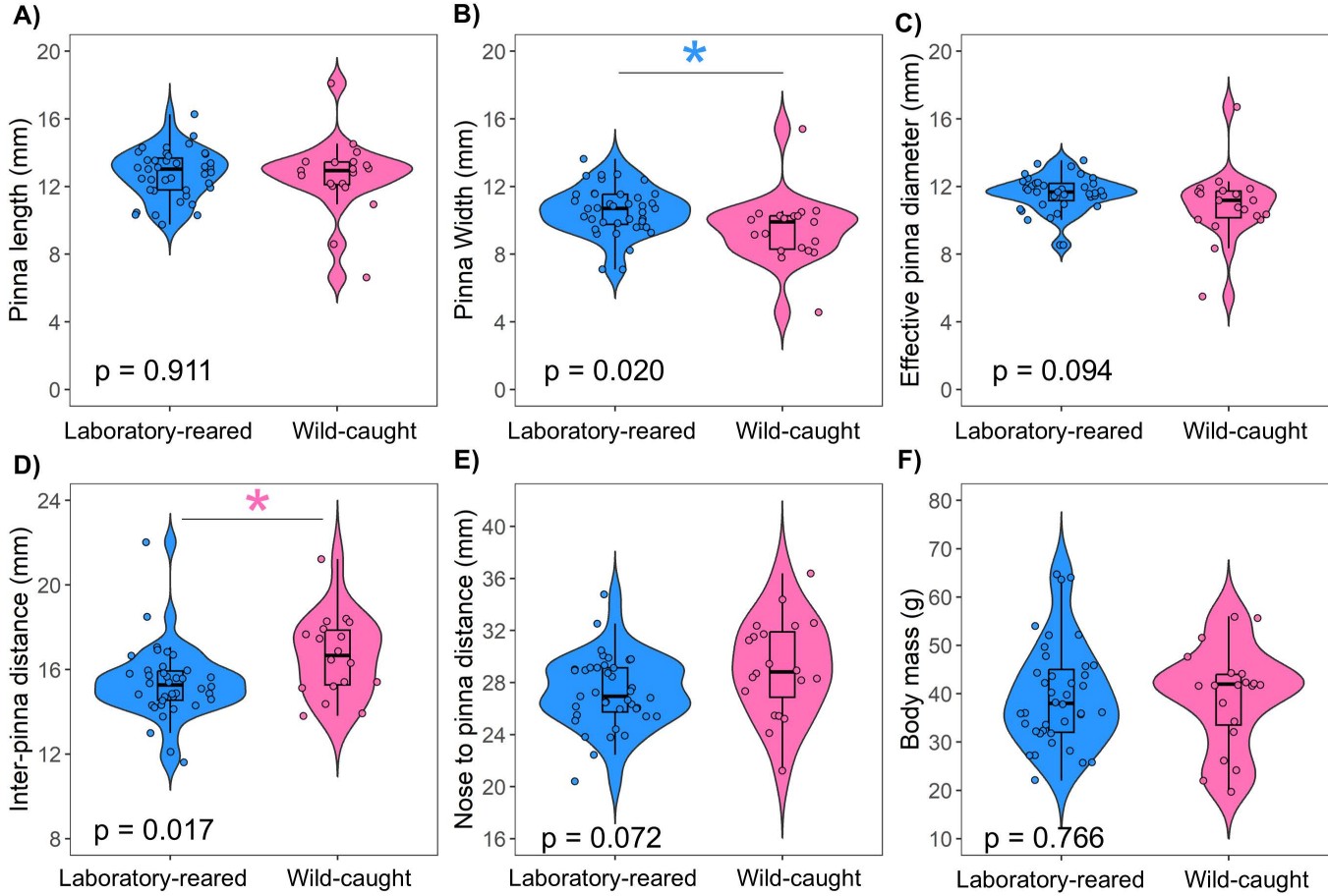

**Fig 7. Morphological measurements between laboratory-reared and wild-caught prairie voles (blue violin plot = laboratory-reared, pink violin plot = wild-caught).** Significant differences were detected on pinna width (B) and inter-pinna distance **(D)**, but not on pinna length **(A)**, effective pinna diameter **(C)**, nose-to-pinna distance **(E)**, and body mass **(F)**.

threshold differences between wild-caught and laboratory-reared prairie voles. Differences in auditory sensitivity are typically associated with noise exposure, age, stress, and hormonal differences. For instance, mice exposed to chronic noise from individually ventilated cages (IVCs) have been reported to have damaged hearing [64,65]. Studies in humans have also demonstrated that exposure to audible noise, especially at high intensities, can influence various physiological and behavioral traits, such as hearing sensitivity, anxiety, and cardiovascular function [66,67]. Since ultrasonic frequencies are audible to rodents [68], it is reasonable to infer that exposure to IVCs noise (high and low frequency) may produce comparable effects to our laboratory-reared individuals.

In addition, age can also influence auditory thresholds, older individuals typically exhibit greater threshold deterioration at frequencies above 4 kHz [13]. While the precise ages of our wild-caught voles could not be determined, published literature indicates that adult individuals (≥ 30 g) are typically > 55 days old (male) or > 45 days old (female), whereas subadult individuals (21–29 g) are generally about 35–45 days old [69]. These estimates suggest that our wild-caught sample likely consisted of young adults and older subadults, which may introduce some variability in threshold measures. However, given that both groups fall within similar developmental stages, we expected age-related effects to be minimal, though not statistically testable. It remains unclear if the differences in ABR thresholds observed in the current investigation reflect differences in environmental ambient noises or developmental effects associated with early acoustic experience, genetic

divergence between wild and laboratory-reared prairie voles, or other environmental and physiological factors. Additional studies are needed to determine whether the differences persist across an animal's lifespan or vary with additional variables such as age or habitat complexity.

ABR frequency response relationships have shown that exclusively subterranean rodents such as naked mole-rats (*Heterocephalus glaber*), coruros (*Spalacopus cyanus*), and Mashona mole-rats (*Fukomys darlingi*) have enhanced sensitivity to low frequencies, with peak sensitivity occurring in the 1–4 kHz region [70]. At 2 kHz, the lowest mean hearing thresholds are approximately 20 dB SPL for coruros, 55 dB SPL for Mashona mole-rats, and 40 dB SPL for naked mole-rats [70]. Our results are consistent with Caspar and colleagues, with both laboratory-reared (49.5 dB SPL) and wild caught (45.5 dB SPL) prairie voles showing mean hearing thresholds around 50 dB SPL at 2 kHz, suggesting that prairie voles, as a semi-fossorial rodent, also exhibit relatively good sensitivity to low frequency sounds. It is important to note that auditory thresholds detected under anesthesia generally tend to be higher than those detected behaviorally, and there is variation between approaches. Studies exploring hearing thresholds between anesthetized and awake subjects suggest that ABR thresholds are generally 10–20 dB higher in anesthetized subjects than those detected behaviorally in awake subjects [71,72]. However, ABR allows for detection of hearing sensitivity in animals without behavioral training. Importantly, it is thought that most small-headed mammals are not capable of generating functional, useful directional information from low-frequency sounds for which timing and level differences for binaural processing are minimal [50,73]. Small-headed mammals (except for some subterranean rodents like the naked mole rat, blind mole rat, and pocket gopher that lack high-frequency hearing) are thus reliant on high-frequency sounds for directional hearing [50,73]. As a result, limited low-frequency sensitivity in small-headed rodents is thought to reflect evolutionary pressures related to their inability to use interaural information for localizing sounds along the horizontal plane [73,74]. Given this framework, the similar low-frequency sensitivity found in wild-caught and lab-reared prairie voles in the current study compared to subterranean rodents is surprising and could reflect their semi-fossorial nature or other means of low frequency cues for communication or other means that are yet unexamined.

Acoustic communication is widespread among rodent species and plays a crucial role in mediating biologically important behaviors, including territorial defense, predator avoidance, courtship, group cohesion, alarm calling, and mother-pup interactions. Laboratory-reared prairie voles emit a total of 14 distinct types of calls, with significant sex differences in both the structure and context of these calls [38,40,75]. However, the structure and function of vocalizations in wild prairie vole populations remain poorly understood. Prairie vole vocalizations span a frequency range of approximately 2.5 to 50 kHz, with the majority of acoustic energy concentrated around 25 kHz [38]. This vocal range aligns with their peak hearing sensitivity, which ranged from 8 to 32 kHz, suggesting that prairie voles are well adapted to detect and discriminate conspecific vocal signals. Notably, wild-caught prairie voles hear better across tested frequencies and exhibit approximately 10 dB SPL lower mean thresholds than lab-reared individuals at 4, 8, 16, and 24 kHz (see Table 1). This enhanced low-frequency sensitivity in wild-caught voles may reflect natural selective pressures for detecting socially relevant signals in more complex or noisy environments and could facilitate more effective communication over long distances. These findings support our hypothesis that wild-caught prairie voles will exhibit lower ABR thresholds than laboratory-reared voles across tested frequencies.

Most studies investigating ABR characteristics in rodent models for auditory research have primarily focused on laboratory-reared individuals. These studies have demonstrated species-specific differences in ABR measures, including variation in click thresholds, as well as differences in monaural and binaural wave amplitudes and latencies [13,46,47,76]. However, relatively few studies have examined wild rodents to assess potential differences in auditory processing between wild-caught and laboratory-reared animals. In our study, we detected significant differences in click thresholds between both tested vole groups, with wild-caught voles exhibiting an average threshold of 52 dB SPL compared to 67 dB SPL in their laboratory-reared counterparts. It is important to highlight that our click threshold detection technique, based on in-ear calibration, is estimated to be less sensitive than actual thresholds by approximately 10–20 dB SPL. Therefore,

the true click thresholds for both wild and laboratory-reared prairie voles may be lower than reported here. We found significant differences in absolute latency of wave IV between wild and laboratory-reared prairie voles, but not at subsequent ABR waves. Similarly, we noticed differences in interpeak wave III − I and wave IV − I. These latency differences, which reflect neural conduction time along the ascending auditory pathway, were shorter in laboratory-reared voles. In contrast, wild-caught prairie voles exhibited significantly higher absolute amplitude for wave I-III (but not wave IV) than laboratory-reared voles. However, there were no differences in amplitude ratios (II/I, III/I, and IV/I) between both groups. Since amplitude ratios generally reflect the gain of brainstem nuclei relative to peripheral auditory input [77,78], these findings suggest that wild-caught and laboratory-reared prairie voles may exhibit comparable levels of central auditory pathway amplification. However, further studies are needed to directly assess the mechanisms underlying this similarity.

All ABR recordings in the present study were performed under anesthesia, which is known to influence auditory physiology in small mammals. Previous studies indicated that commonly anesthetic injection can decrease absolute peak amplitudes and increase peak latencies [6,51]. While these effects generally apply across species consistently, they may alter the absolute values reported here relative to those obtained from awake animals. Nevertheless, we expected that the relative differences we observed among tested groups remain meaningful despite anesthesia-related shifts in absolute response magnitude. Future studies incorporating awake auditory recordings or minimally invasive electrophysiological techniques could shed light on the degree to which neural responses in these groups differ across anesthetized and awake states.

Although this study is the first to compare binaural sound processing between laboratory and wild-caught prairie voles, our findings align with previous publications showing how the BIC varies with ITD in rodents [47,51,56,79]. Small-headed mammals use interaural level difference (ILD) and ITD information for localizing sound sources, and due to their small head size, rodents typically display a smaller range of ITD cues compared to other species (such as *Felis catus* or *Chinchilla lanigera*) that are known to use ITD cues [56]. Consistent with previous studies, this work shows a positive relationship of normalized latency with increasing ITD and a corresponding decrease in BIC relative amplitude as ITD increases [46,47,50,51,56]. We found faster normalized DN1 latency at 1.5 and 2.0 ms and higher relative DN1 amplitude of the BIC across tested ITDs between groups. Since the morphometric measurements (except pinna width and inter-pinna distance) data were not statistically significant between groups, we suggest that craniofacial and pinna features did not account for the differences in binaural sound processing observed here. However, differences in auditory processing may still arise from other factors not dependent on external pinna and craniofacial morphology. For instance, early exposure to environmental noise and diverse social interactions in the wild could drive the development of the auditory system and neural responsiveness through-experience [20,21,80]. In addition, animals that inhabit natural environments are likely to encounter more diverse and behaviorally relevant acoustic stimuli, which natural selection may favor fine-tuned narrow central auditory processing [21]. Conversely, wild animals may face challenges such as pathogen exposure, parasitic infections, nutritional stress, or injuries that could compromise auditory function or increase variability in performance relative to their laboratory-reared counterparts [81]. It is therefore not surprising that laboratory-reared prairie voles exhibit better binaural hearing sensitivity than their wild-caught counterparts. Broader comparative studies across multiple wild and laboratory-reared species would be necessary to confirm what is reported here.

Sex is an important variable to consider when performing auditory processing research. Indeed, previous studies have demonstrated sex differences in ABR wave characteristics in rodents, with females typically displaying significantly lower click and frequency ABR thresholds, and exhibiting higher amplitude and faster latency of ABR wave II and IV than males [50,51]. In this study, no sex differences were detected in overall ABR frequency and click thresholds between prairie vole groups. Likewise, there were no significant effects of sex on interpeak latency, DN1 latency, amplitude ratio, and relative amplitude between groups. However, within group comparisons revealed differences in auditory thresholds and ABR wave characteristics. For instance, female wild-caught voles demonstrated better hearing sensitivity than female laboratory reared voles at 1, 4, 8 kHz. In addition, female laboratory-reared voles exhibited faster normalized latencies than female

wild-caught voles at 2.0 ms ITD. Similar trends were observed between male wild-caught and male lab-reared prairie voles. It is difficult to speculate on the underlying causes leading to these sex differences within groups. One plausible explanation could be age-related variation between individuals within sexes. Mating status may also play a role, as physiological and hormonal differences between mated and unmated individuals can influence sensory processing, including frequency auditory thresholds [76]. Other contributing factors may be early acoustic exposure, genetic diversity, or stress. Further studies that control for age, mating status, and developmental environment are needed to clarify the mechanism driving these effects observed within sexes in this study.

We compared morphometric features such as pinna width, pinna length, nose to pinna distance, effective pinna diameter, body mass, and body length between wild-caught and laboratory-reared prairie voles. We found no significant differences in any of the craniofacial and pinna characteristic features (except pinna width and inter-pinna distance) between the two groups. Previous studies have reported that adult wild prairie voles typically weigh approximately 35 g, while laboratory-reared individuals average around 35–40 g [14,35], which was consistent with our findings. In addition, it has been suggested that the Kansas prairie vole population exhibits some degree of sexual dimorphism in body mass, with males tending to be heavier than females [82]. However, in the present study, we did not observe significant differences in body mass between the sexes in the Kansas population. This discrepancy may reflect population-specific variation, differences in sampling season or age structure. Our results suggest no sexual dimorphism with respect to body mass in the Lawrence, Kansas population and indicate that patterns of sexual dimorphism in prairie voles in Kansas may not be consistent across populations. Finally, no previous research has reported craniofacial and pinna features measurement comparisons between wild and laboratory voles. This study provides detailed measurements of craniofacial and pinna dimensions for both groups, which could serve as a valuable reference for future research on auditory morphology and its potential relationship to hearing sensitivity or acoustic communication in this species.

Although we revealed differences in auditory processing between laboratory and wild prairie voles, the precise mechanism(s) that drive these differences is not known and necessitate further investigation. Nonetheless, it is possible that the differences in hearing sensitivity between laboratory and wild prairie voles may be either a result of domestication, anesthetic used, or selective pressure from natural environments. It has been suggested that domestication may alter behavioral traits in animals across generations [83], and in small mammals such as rodents, captivity has led to reduced variability in aggression, exploratory behavior, locomotion activity, morphology features, and reproduction [84–88]. Generations in domestication may therefore reduce prairie voles' ability to hear as well as wild-caught voles. However, precise generational data limit our ability to quantify the extent to which long-term captive breeding may have contributed to the observed auditory difference and therefore is a limitation to this study. Regardless of the mechanism, a difference exists and more investigations are needed to comprehend why we observe the differences in ABR thresholds, click thresholds, monaural and binaural latencies and amplitudes between laboratory and wild-caught prairie voles. Specifically, further auditory studies using wild animals will help facilitate our understanding of the ultimate reasons and proximate mechanisms that account for the differences in auditory processing and sound localization ability of rodents. Efforts to comprehend differences in auditory processing between wild and lab prairie voles are critical due to the extensive use of prairie voles as model organisms in social neuroscience studies as well as a better understanding of laboratory-reared conditions that may be affecting hearing in other species.

## Conclusion

This study offers novel insights into auditory processing between laboratory and wild prairie voles. By using ABR recording measurements, we show that wild-caught and laboratory-reared prairie voles exhibit a hearing range from 1–46 kHz, with lowest thresholds between 8–32 kHz. Wild-caught voles displayed significantly lower ABR thresholds at 1, 4, 8, 16, and 24 kHz as well as lower click thresholds than laboratory-reared voles. Similarly, we report differences in monaural and binaural hearing in this species, with laboratory-reared voles showing faster monaural and binaural responses compared

to their wild-caught counterparts. Future studies incorporating behavioral and physiological assessments, particularly examining the effects of age on auditory function and vocalizations, are essential to determine whether the observed differences in auditory processing are influenced by age-related factors in both wild-caught and laboratory prairie vole populations.

## Supporting information

**S1 File. Electronic Supplemental Material.** Tables supporting audiogram data for males and females.
(DOCX)

## Acknowledgments

We are grateful to Marie Stone and Sheena Parsons for housing and authorizing us to trap rodents at the Selman Living laboratory and the Kansas University Field Station. We are also grateful to Dr. Tim Lei and Dr. Ben-Zheng Li for creating the custom Python tools for ABR data acquisition and analysis. We thank Dr. Tom Curtis for generously sharing his laboratory-reared prairie vole colony and for his assistance in establishing and maintaining our colony.

## Author contributions

**Conceptualization:** Luberson Joseph, Jessica A. Hurd, Casey E. Sergott.

**Data curation:** Luberson Joseph, Jessica A. Hurd, Casey E. Sergott, Elizabeth A. McCullagh.

**Formal analysis:** Luberson Joseph, Jessica A. Hurd, Casey E. Sergott.

**Investigation:** Luberson Joseph, Naleyshka Colon-Rivera, Emily M. New, Desi M. Joseph, Jessica A. Hurd, Casey E. Sergott, Elizabeth A. McCullagh.

**Methodology:** Luberson Joseph, Naleyshka Colon-Rivera, Emily M. New, Desi M. Joseph, Jessica A. Hurd, Casey E. Sergott.

**Project administration:** Luberson Joseph, Elizabeth A. McCullagh.

**Resources:** Elizabeth A. McCullagh.

**Supervision:** Elizabeth A. McCullagh.

**Validation:** Luberson Joseph, Naleyshka Colon-Rivera, Elizabeth A. McCullagh.

**Visualization:** Luberson Joseph, Desi M. Joseph.

**Writing – original draft:** Luberson Joseph, Naleyshka Colon-Rivera, Jessica A. Hurd, Casey E. Sergott.

**Writing – review & editing:** Luberson Joseph, Naleyshka Colon-Rivera, Emily M. New, Desi M. Joseph, Jessica A. Hurd, Casey E. Sergott, Elizabeth A. McCullagh.

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
