## [Decision Letter · Decision Letter 0]

26 Nov 2025

Dear Dr. McCullagh,

Thank you for submitting your manuscript to PLOS ONE. After careful consideration, we feel that it has merit but does not fully meet PLOS ONE’s publication criteria as it currently stands. Therefore, we invite you to submit a revised version of the manuscript that addresses the points raised during the review process.

Please add a more deep discusion about ABR differences between mammals in awake and anesthetized conditions

We look forward to receiving your revised manuscript.

Kind regards,

Paul H Delano, Ph.D.

Academic Editor

PLOS ONE

Journal Requirements:

2. To comply with PLOS One submissions requirements, in your Methods section, please provide additional information regarding the experiments involving animals and ensure you have included details on (1) methods of sacrifice, and (2) efforts to alleviate suffering.

Reviewers' comments:

Reviewer's Responses to Questions

**Comments to the Author**

1. Is the manuscript technically sound, and do the data support the conclusions?

Reviewer #1: Yes

Reviewer #2: Yes

2. Has the statistical analysis been performed appropriately and rigorously?

Reviewer #1: Yes

Reviewer #2: I Don't Know

3. Have the authors made all data underlying the findings in their manuscript fully available?

Reviewer #1: Yes

Reviewer #2: No

4. Is the manuscript presented in an intelligible fashion and written in standard English?

Reviewer #1: Yes

Reviewer #2: Yes

Reviewer #1: The work carried out by the authors is well written and methodologically sound. I believe that having information on the electrophysiological values and binaural measurements of both rodent strains is a valuable contribution to future studies on hearing or vocalizations.

As a general comment, I would ask the authors to go into a little more depth in their discussion of the work on estimating the age of the rodents (understanding that it was not possible to date the age of the wild strain), since we know that the older the rodents are, the greater the differences in thresholds, latencies, and amplitudes in the ABR. When the existence of groups (adults (≥ 30 g) and subadults (21 to 29 g)) is pointed out, is there any estimate of what the age range of adults and subadults would be? If not, I think this point should also be discussed in a paragraph in the text.

On the other hand, all recordings were made under anesthesia. Could this have influenced the results in any way? Although the ABR is a brainstem recording, it has been reported that in some rodents there may be differences in absolute latency peaks in awake and anesthetized conditions. This study hypothesizes that the results obtained (e.g., increased amplitude or decreased ABR latencies) could be due to domestication or selective pressure from the environment of these rodents, which, in turn, would influence exploratory behaviors or locomotor skills that would help the animals perform better in a natural environment. On the other hand, the behaviors discussed for these animals (wild-caught prairie voles) were performed in natural environments, that is, while awake. That said, is it entirely correct to extrapolate this explanation based on recordings made under anesthesia? Could there be differences in both cases (anesthetized and non-anesthetized)? I believe this should also be discussed in the text in a paragraph and possible limitations in this regard should be mentioned.

Reviewer #2: This study evaluates differences in ABR between laboratory and wild caught prairie voles finding differences between origin but not between sexes. The results may be of interest for the readers of the journal; however I find some issues that should be clarified/corrected before recommending the acceptance.

The introduction needs more work, the order should go from general relevant background to the specific problem that the study will address. There is an entire paragraph without references. More information supporting the idea of differential selection pressures should be included. In addition, there is no clear background supporting the hypothesis regarding the morphological differences. The statistical procedures requires more detailed explanation and there may be an issue if modeling assumptions were not met. In the results section, the statistics values and the degrees of freedom should be provided. I find the results section excessively long, I would recommend trying to find a more concise way to present the results and/or to move some parts to supplementary material. In the discussion section there is also text without references.

L48-59: This paragraph includes several ideas and information that should be referenced.

L61: Please include the scientific name.

L77: Please clarify what means ABR.

L80-91: This paragraph should be placed before presenting and mentioning the characteristics of the species under study.

L96-97: Please add information about the number of generations that the laboratory-reared group have been in captivity conditions. This is needed for a better support for the proposed hypothesis.

L101-103: I find very hard to understand the rationale of this hypothesis. Please provide more background in the previous parts of the introduction.

L184: “… gradually decreasing the stimulus intensity in 10 dB SPL increments”, please check if “increments” is the proper term to be used.

L241: RStudio is an IDE that allows an easier use of R. Please provide the correct references.

L243-248: Why body mass was not included in the models?

L248: Estimated marginal means were obtained with the emmeans package?

L248: Please clarify if the model used for a posteriori analyses was the minimal adequate model.

L250: As far as I know, the lme4 package does not provide p-values. Please clarify the procedures.

L253-254: As the data was not normally distributed, please clarify how the model (LMMs) assumptions were evaluated (e.g. normal distribution of residuals). If assumptions are not met the results may be wrong.

L261: It seems that for wild-caught the range is between 8 and 24 kHz.

L263: Please indicate the statistic value and the degrees of freedom when reporting the significance of the fixed effects.

L267: The values of the statistics and of the p-values are reported in Table 1. I would recommend to not repeat this information.

L660-671: This paragraph includes several ideas and information that should be referenced.

**Do you want your identity to be public for this peer review?** For information about this choice, including consent withdrawal, please see our Privacy Policy

Reviewer #1: **Yes:** Cristian Aedo-Sanchez

Reviewer #2: No

---

## [Author Response · Author response to Decision Letter 1]

16 Dec 2025

Reviewer # 1

The work carried out by the authors is well written and methodologically sound. I believe that having information on the electrophysiological values and binaural measurements of both rodent strains is a valuable contribution to future studies on hearing or vocalizations.

As a general comment, I would ask the authors to go into a little more depth in their discussion of the work on estimating the age of the rodents (understanding that it was not possible to date the age of the wild strain), since we know that the older the rodents are, the greater the differences in thresholds, latencies, and amplitudes in the ABR. When the existence of groups (adults (≥ 30 g) and subadults (21 to 29 g)) is pointed out, is there any estimate of what the age range of adults and subadults would be? If not, I think this point should also be discussed in a paragraph in the text.

Thank you for pointing out the importance of age-related auditory differences. We have now expanded the discussion to provide context regarding age estimation in wild-caught prairie voles. Although exact ages cannot be determined, published literature indicates that prairie voles reach the adult body mass range (≥ 30 g) typically at > 45 (female) and 55 (male) days of age, whereas subadults in the 21 to 29 g range are typically 35 to 45 days old (Mateo et al., 1994). We have added additional information in the discussion acknowledging these estimates and discussing how the limited precision in age determination may influence ABR measures and interpretation.

On the other hand, all recordings were made under anesthesia. Could this have influenced the results in any way? Although the ABR is a brainstem recording, it has been reported that in some rodents there may be differences in absolute latency peaks in awake and anesthetized conditions. This study hypothesizes that the results obtained (e.g., increased amplitude or decreased ABR latencies) could be due to domestication or selective pressure from the environment of these rodents, which, in turn, would influence exploratory behaviors or locomotor skills that would help the animals perform better in a natural environment. On the other hand, the behaviors discussed for these animals (wild-caught prairie voles) were performed in natural environments, that is, while awake. That said, is it entirely correct to extrapolate this explanation based on recordings made under anesthesia? Could there be differences in both cases (anesthetized and non-anesthetized)? I believe this should also be discussed in the text in a paragraph and possible limitations in this regard should be mentioned.

We thank the reviewer for this thoughtful comment. All recordings in this study were performed under anesthesia and we agree that anesthetic injections can influence auditory brainstem responses measurements including increasing auditory thresholds and absolute peak latencies, and reducing response amplitudes in some rodent species. To address this concern, we have updated our discussion to acknowledge the potential influence of anesthesia on our ABR measurements. In addition, we highlight that despite these potential effects, the relative differences among tested groups are likely preserved, but should still be interpreted with caution.

Reviewer # 2

This study evaluates differences in ABR between laboratory and wild caught prairie voles finding differences between origin but not between sexes. The results may be of interest for the readers of the journal; however I find some issues that should be clarified/corrected before recommending the acceptance.

The introduction needs more work, the order should go from general relevant background to the specific problem that the study will address. There is an entire paragraph without references. More information supporting the idea of differential selection pressures should be included. In addition, there is no clear background supporting the hypothesis regarding the morphological differences. The statistical procedures requires more detailed explanation and there may be an issue if modeling assumptions were not met. In the results section, the statistics values and the degrees of freedom should be provided. I find the results section excessively long, I would recommend trying to find a more concise way to present the results and/or to move some parts to supplementary material. In the discussion section there is also text without references.

We thank the reviewer for their careful evaluation of our manuscript and for the constructive comments. We have revised the manuscript extensively to address each concern raised. Specifically, we have reorganized and expanded the introduction to provide a clearer progression from general to specific, added appropriate citations where references were previously missing, and incorporated additional literature supporting our morphological hypothesis. We have also clarified the statistical procedures, explicitly describing how model assumptions were evaluated, and revised the results to include test statistic and degrees of freedom. We carefully consider the reviewer’s suggestion regarding furthering condense the results section.. However, we choose to retain the current structure of the results to facilitate reader interpretation and to clearly present within group and between groups differences that are central to the study. Finally, the discussion was revised to include additional references where appropriate and to better contextualize our findings. We believe these revisions substantially strengthen the manuscript.

Finally, the Discussion was revised to include additional references where appropriate and to better contextualize our findings within the existing literature. We believe these revisions substantially strengthen the manuscript and address all of the reviewer’s concerns.

L48-59: This paragraph includes several ideas and information that should be referenced.

Thank you for highlighting this, we have added references to this paragraph in the final manuscript.

L61: Please include the scientific name.

We have included the scientific name of the prairie voles (Microtus ochrogaster) in the final manuscript.

L77: Please clarify what means ABR.

We thank the reviewer for this comment. The abbreviation ABR (Auditory Brainstem Response) is already defined as its first occurrence in the abstract of the manuscript, in accordance with journal guidelines. For this reason, we did not repeat the definition in the main text.

L80-91: This paragraph should be placed before presenting and mentioning the characteristics of the species under study.

We have moved this paragraph before the paragraph introducing the characteristics of the studied species.

L96-97: Please add information about the number of generations that the laboratory-reared group have been in captivity conditions. This is needed for a better support for the proposed hypothesis.

We thank the reviewer for this comment. Precise information regarding the exact number of generations that the laboratory-reared voles have been maintained in captivity is not available. However, we have now added clarifying information regarding the origin of the laboratory colony. Specifically, the laboratory-reared voles trace back to colonies originally established from wild-caught prairie voles in Illinois in the 1980s and have since been maintained through extensive outcrossing among multiple university laboratory colonies. As a result, while these animals have a long history in captive breeding, the majority of their genetic background originates from wild population, and the exact number of captive generations cannot be reliably determined. We have included this clarification in the methods and acknowledged this limitation in the discussion of the final manuscript.

L101-103: I find it very hard to understand the rationale of this hypothesis. Please provide more background in the previous parts of the introduction.

We thank the reviewer for this comment. To clarify and strengthen the rationale for this hypothesis, we have expanded the introduction to provide additional background on sexual dimorphism in prairie voles. Specifically, we now summarize evidence showing that, despite documented population level variation in mating system and social behavior of prairie voles from Tennessee, Illinois, and Kansas, prairie voles generally exhibit minimal and inconsistent sexual dimorphism in external morphological features, and prior reports of increased dimorphism remain equivocal. This literature base context provides a clearer foundation for our hypothesis that morphological measurements would not differ between wild-caught and laboratory-reared prairie voles.

L184: “… gradually decreasing the stimulus intensity in 10 dB SPL increments”, please check if “increments” is the proper term to be used.

We have replaced increments by steps in the revised manuscript.

L241: RStudio is an IDE that allows an easier use of R. Please provide the correct references.

We thank the reviewer for this suggestion and have added the appropriate reference for R in the revised manuscript.

L243-248: Why body mass was not included in the models?

We thank the reviewer for this comment. Body mass was not included in the models because there was no difference in body mass in our dataset. Additionally, exploratory analyses including body mass in our model did not improve the best fit model.

L248: Estimated marginal means were obtained with the emmeans package?

Yes, we used the emmeans package to obtain marginal means. We clarified this in the methods section of the manuscript.

L248: Please clarify if the model used for a posteriori analyses was the minimal adequate model.

We thank the reviewer for this comment. Minimal adequate model selection was explicitly conducted prior to all a posteriori analyses. We compared the null model, the additive model, and the full model using a likelihood ratio test to get minimal adequate model. All post hoc comparisons were conducted on the minimal adequate model.

L250: As far as I know, the lme4 package does not provide p-values. Please clarify the procedures.

We thank the reviewer for this comment. While the core lme4 package does not provide p-values for fixed effects by default. All linear mixed-effects models were fitted using the lmerTest package, which extends lme4. We have clarified this procedure in the methods section.

L253-254: As the data was not normally distributed, please clarify how the model (LMMs) assumptions were evaluated (e.g. normal distribution of residuals). If assumptions are not met the results may be wrong.

We thank the reviewer for this comment. LMMs assume that model residuals are approximately normally distributed. Model assumptions were evaluated by visual inspection of residual diagnostic plots, including Q-Q plots and residuals versus fitted values for all final models. No substantial deviations from normality or homoscedasticity were observed. We have clarified this procedure in the methods section.

L261: It seems that for wild-caught the range is between 8 and 24 kHz.

We have updated this information in the revised version of the manuscript.

L263: Please indicate the statistic value and the degrees of freedom when reporting the significance of the fixed effects.

We have revised the results and provided these values to the revised manuscript.

L267: The values of the statistics and of the p-values are reported in Table 1. I would recommend to not repeat this information.

We have removed the statistics and p-values in the text and referred to table 1.

L660-671: This paragraph includes several ideas and information that should be referenced.

We have added additional references to this section of the discussion in the revised manuscript.

---

## [Decision Letter · Decision Letter 1]

22 Jan 2026

Differences in Auditory Brainstem Responses between Laboratory-reared and Wild-caught Prairie Voles (Microtus ochrogaster)

PONE-D-25-54222R1

Dear Dr. McCullagh,

We’re pleased to inform you that your manuscript has been judged scientifically suitable for publication and will be formally accepted for publication once it meets all outstanding technical requirements.

Kind regards,

Paul H Delano, Ph.D.

Academic Editor

PLOS One

Additional Editor Comments (optional):

Reviewers' comments:

Reviewer's Responses to Questions

**Comments to the Author**

Reviewer #1: All comments have been addressed

Reviewer #2: (No Response)

2. Is the manuscript technically sound, and do the data support the conclusions?

Reviewer #1: Yes

Reviewer #2: Yes

3. Has the statistical analysis been performed appropriately and rigorously?

Reviewer #1: Yes

Reviewer #2: Yes

4. Have the authors made all data underlying the findings in their manuscript fully available?

Reviewer #1: Yes

Reviewer #2: Yes

5. Is the manuscript presented in an intelligible fashion and written in standard English?

Reviewer #1: Yes

Reviewer #2: Yes

Reviewer #1: In this second version of the paper, the authors have responded to and discussed the requirements regarding the analysis of the animals' age and the influence of anesthesia on ABR recordings in an appropriate manner.

I have no further requirements or additional comments for the authors.

Congratulations on the work you have done.

Reviewer #2: (No Response)

**Do you want your identity to be public for this peer review?** For information about this choice, including consent withdrawal, please see our Privacy Policy

Reviewer #1: **Yes:** Cristian Aedo Sanchez

Reviewer #2: No

---

## [Editor Report · Acceptance letter]

PONE-D-25-54222R1

PLOS One

Dear Dr. McCullagh,

I'm pleased to inform you that your manuscript has been deemed suitable for publication in PLOS One. Congratulations! Your manuscript is now being handed over to our production team.

Kind regards,

on behalf of

Dr. Paul H Delano

Academic Editor

PLOS One